# Mechanism for priming DNA synthesis by yeast DNA Polymerase α

**Rajika L Perera[1], Rubben Torella[2], Sebastian Klinge[1†], Mairi L Kilkenny[1], Joseph D Maman[1], Luca Pellegrini[1]***

[1]Department of Biochemistry, University of Cambridge, Cambridge, United Kingdom; [2]Unilever Centre of Molecular Informatics, Department of Chemistry, University of Cambridge, Cambridge, United Kingdom

**Abstract** The DNA Polymerase α (Pol α)/primase complex initiates DNA synthesis in eukaryotic replication. In the complex, Pol α and primase cooperate in the production of RNA-DNA oligonucleotides that prime synthesis of new DNA. Here we report crystal structures of the catalytic core of yeast Pol α in unliganded form, bound to an RNA primer/DNA template and extending an RNA primer with deoxynucleotides. We combine the structural analysis with biochemical and computational data to demonstrate that Pol α specifically recognizes the A-form RNA/DNA helix and that the ensuing synthesis of B-form DNA terminates primer synthesis. The spontaneous release of the completed RNA-DNA primer by the Pol α/primase complex simplifies current models of primer transfer to leading- and lagging strand polymerases. The proposed mechanism of nucleotide polymerization by Pol α might contribute to genomic stability by limiting the amount of inaccurate DNA to be corrected at the start of each Okazaki fragment.

***For correspondence:** lp212@cam.ac.uk

[†]**Present address:** Institut für Molekularbiologie und Biophysik, Eidgenössische Technische Hochschule Hönggerberg, Zurich, Switzerland

**Competing interests:** The authors have declared that no competing interests exist

**Reviewing editor**: John Kuriyan, University of California, Berkeley, United States

## Introduction

Cellular organisms initiate DNA synthesis during genome duplication by the universal mechanism of RNA priming, the assembly of short RNA molecules on the unwound strands of the DNA helix by a specialized DNA-dependent RNA polymerase known as primase (*Frick and Richardson, 2001*; *Kuchta and Stengel, 2010*; *DePamphilis and Bell, 2011*). The RNA primers are extended in an obligate 5′ to 3′ direction by the replicative DNA polymerases that synthesize the bulk of chromosomal DNA. The initiation of DNA synthesis is made more complicated by the concurrent duplication of the antiparallel strands of parental DNA (*Hamdan and van Oijen, 2010*). The repeated priming events necessary for the discontinuous synthesis of the lagging strand require the constant activity of primase at the replication fork.

In bacteria and bacteriophages, RNA priming is performed by a single-chain primase, acting in combination with the replicative helicase, to which it is bound by a dynamic interaction or is fused together in the same polypeptide (*Patel et al., 2011*). The molecular apparatus responsible for initiation of DNA synthesis in eukaryotic replication is more complex. The eukaryotic primase is a heterodimer of catalytic and regulatory subunits that is associated with the catalytic subunit of Pol α and its accessory B subunit in a constitutive heterotetrameric assembly, the Pol α/primase complex (*Loeb et al., 1986*; *Kaguni and Lehman, 1988*; *Foiani et al., 1997*; *Lao-Sirieix et al., 2005b*). Reflecting its critical importance to DNA replication, the Pol α/primase complex is an integral component of the eukaryotic replisome (*Calzada et al., 2005*).

The oligonucleotides synthesized by bacterial and bacteriophage primases are between 4 and 12 nucleotides long and made exclusively of RNA. In contrast, the Pol α/primase complex produces longer, composite RNA-DNA primers that result from the concerted enzymatic activities of primase and Pol α (*Chang et al., 1984*; *Hu et al., 1984*; *Singh, et al., 1986*). The constitutive association of Pol

**eLife digest** During mitosis, a cell duplicates its DNA and then divides, ultimately generating two genetically identical daughter cells. In eukaryotes, the process of DNA duplication occurs at multiple sites throughout the genome: at each site, the antiparallel strands of the parental DNA separate and provide a template for DNA polymerase (Pol), the enzyme that synthesizes the two new DNA strands. Duplication of the DNA proceeds in both directions from each site through the polymerization of nucleotides to form new strands of DNA that are complementary to the template strands. However, since DNA polymerases can only polymerize nucleotides in one direction, the 5′ to 3′ direction, synthesis of the so-called leading strand proceeds continuously, whereas the other, lagging strand is synthesized in fragments.

The task of duplicating the bulk of the DNA is shared between Pol δ, which is primarily responsible for synthesis of the lagging strand, and Pol ε, which fulfils the same role for the leading strand. However, Pols δ and ε cannot initiate DNA synthesis by themselves; short RNA-DNA chains called primers must also be paired to each template strand. Production of the primers requires the concerted action of two more enzymes: an RNA polymerase known as primase, and another DNA polymerase called Pol α. It is known that completion of the RNA-DNA primer requires Pol α to increase the length of the RNA segment by adding extra nucleotides, but the details of this process are poorly understood.

Perera et al. combined crystallographic, biochemical and computational evidence to describe how Pol α first recognizes and then extends the RNA strand in the primer. They found that Pol α recognizes the particular shape of double helix—an A-form helix—that is formed by the DNA template and the RNA primer. The geometry of this helix prompts the Pol α enzyme to start adding nucleotides to the RNA in the primer. Perera et al. determined that once a full turn of double-helix DNA has been synthesized, Pol α is no longer in direct contact with the A-form helix, which causes the enzyme to disengage and terminate polymerization, leaving behind the now complete RNA-DNA primer.

Perera et al. offer a new paradigm for understanding the initiation of DNA synthesis in eukaryotic replication. Their work suggests that Pol α has the ability to discriminate between different shapes of the primer-template helix, thus providing a mechanistic understanding of primer release. The spontaneous release of the primer offers a simple and elegant way to limit DNA synthesis by Pol α, a polymerase that is prone to error, and to make the RNA-DNA primer directly available for extension by Pol δ and Pol ε.

α and primase in the cell reflects presumably their tight functional coordination, demanded by the frequent priming necessary for lagging strand synthesis. Detailed knowledge of the mechanism of primer synthesis is lacking, but it must involves initiation, extension and completion of RNA synthesis by primase, intramolecular hand-off of the RNA oligonucleotide to Pol α and limited RNA extension with deoxynucleotides (dNTPs) (*Brooks and Dumas, 1989*; *Kuchta et al., 1990*; *Copeland and Wang, 1993*; *Sheaff and Kuchta, 1993*; *Sheaff et al., 1994*). The large subunit of primase performs a critical function in the primer initiation step via its Fe-S domain (*Klinge et al., 2007*; *Weiner et al., 2007*), whereas the B subunit of Pol α lacks enzymatic activity and likely acts as a scaffold to mediate interactions with other components of the replicative apparatus (*Uchiyama and Wang, 2004*; *Klinge et al., 2009*; *Zhou et al., 2012*).

After completion of primer synthesis by Pol α/primase, the primer is elongated by the processive Pols δ and ε that synthesize the majority of chromosomal DNA on the lagging and leading strand templates, respectively. Synchronization of priming with Okazaki fragment synthesis requires a concerted mode of primer transfer from Pol α to Pol δ and several mechanisms of polymerase switch have been put forward (*Yuzhakov et al., 1999*; *Maga et al., 2000*). Before ligation of the completed Okazaki fragments, the RNA portion of the primer is excised by specific nucleases such as Fen1 and Dna2 (*Burgers, 2009*). The composite nature of the RNA-DNA primers poses a special challenge to the eukaryotic replication apparatus, that must further correct the DNA segment synthesized by Pol α, which lacks proofreading activity; current evidence indicates that the DNA portion of the primer might be corrected by Pol δ (*Pavlov et al., 2006*).

Despite its central role in genomic duplication, the molecular mechanism of primer synthesis by the Pol α/primase complex is poorly understood and structural insights remain limited. Here we use a multi-disciplinary approach to elucidate the structural basis for the catalytic role of Pol α in the synthesis of the RNA-DNA oligonucleotides that prime DNA synthesis in eukaryotic replication. We demonstrate that Pol α recognizes the intrinsic and induced conformation of the A-form RNA primer/DNA template helix and that the resulting synthesis of B-form DNA forms the basis for a feedback mechanism of primer termination. Our findings provide a new paradigm for a critical step in the complex choreography of events that ultimately lead to replication of the lagging DNA strand.

## Results

### Overall structure of actively copying Pol α

We have determined crystal structures of the catalytic core (349–1258; 910 amino acids) of yeast Pol α in unliganded form (apo), bound to an RNA primer/DNA template duplex (binary complex) and in a productive complex with RNA primer/DNA template and incoming dGTP (ternary complex). For structural studies of the ternary complex, we used a polymerase mutant (D998N) with attenuated catalytic activity. During crystallization of the ternary complex, the polymerase extended the 3′-end of the RNA primer by addition of two deoxyguanosine nucleotides. Thus, our crystal structure captures the actively copying Pol α in the act of extending the RNA primer with deoxynucleotides (*Figure 1A* and *Figure 1—figure supplement 1–4*, *Table 1*).

The catalytic region of Pol α adopts the universal 'right-hand' DNA polymerase fold consisting of a palm domain harboring the active site, a fingers domain that interacts with the incoming nucleotide and a thumb domain that grips the primer-template duplex. As the prototypical member of the B-family of DNA polymerases, distinctive structural features that had been identified previously in bacteriophage RB69 Pol (*Wang et al., 1997*), bacterial DNA Pol II (*Wang and Yang, 2009*) and yeast Pol δ (*Swan et al., 2009*), such as the extended N-terminal region or the antiparallel hairpin fold of the helical fingers domain, are also present in Pol α.

The majority of the contacts of the polymerase with the primer/template duplex takes place within a region of 7 bp from the 3′-terminus of the templated primer (*Figure 1B*). Pol α's footprint on the primer-template duplex is smaller than normally observed in B-family DNA polymerases and matches the minimal primer size that is efficiently utilized by the polymerase (*Kuchta et al., 1990*). Four deoxy-nucleotides of single-stranded DNA template fit in extended conformation within the groove formed by the exonuclease domain and N-terminal region of Pol α, in agreement with the position of template DNA upstream of the active site previously observed in other B-family DNA polymerases (*Hogg et al., 2007*; *Swan, et al., 2009*; *Figure 1—figure supplement 5*).

In the crystals, two ternary complexes are related by non-crystallographic dyad symmetry, which brings into contact one end of each RNA/DNA duplex with the palm domain of the other polymerase molecule (*Figure 1—figure supplements 6 and 7*). The acidic tip of a long loop connecting the palm and finger domains unwinds the first base pair of the RNA/DNA duplex, capturing the 5′-end nucleotide of the RNA primer in a surface pocket on the palm domain. Such interaction would potentially begin the unwinding of the templated RNA primer, in preparation for its excision. However, the physiological relevance of our structural observation is currently unclear.

The RNA primer–DNA template helix bound to Pol α adopts overall an A-DNA conformation. Clustering analysis for each dinucleotide step of the mean z-coordinates of the phosphorus atoms with the glycosyl torsion angle χ (*Lu et al., 2000*) identifies a continuous stretch of six dinucleotide steps as A-DNA and one step as B-DNA flanked by dinucleotides in A-like conformation (*Figure 1C*). As a control, the clustering analysis identifies correctly the double-stranded DNA bound productively by B-family Pol δ and RB69 Pol as B-form DNA and the dinucleotide step at the primer 3′-end as of A-like character.

### Specific recognition of the RNA primer

Extensive structural analysis has demonstrated that the palm and thumb domains of actively copying DNA polymerases make continuous contact with the minor groove of the primer-template DNA over a full turn of the DNA double helix. Interactions of the palm domain with the primer-template help position the 3′-terminus of the primer in optimal arrangement for catalysis, whereas the thumb domain secures the grip of the polymerase onto the DNA duplex. Strikingly, the structure of Pol α bound

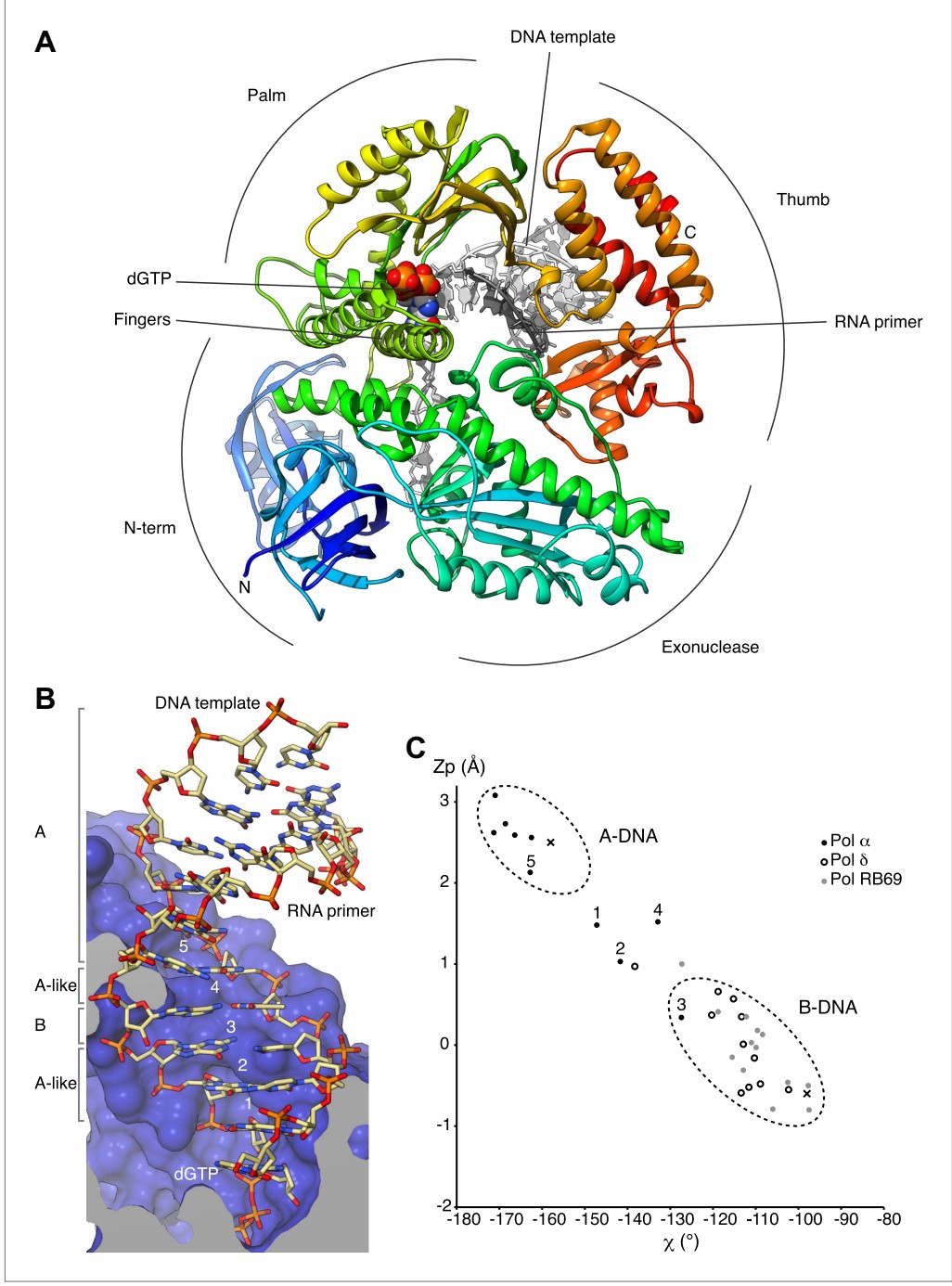

**Figure 1**. Overall structure of actively copying Pol α. (**A**) The polymerase domain of yeast Pol α is shown in complex with an RNA primer/DNA template and dGTP bound in the active site. The polymerase is represented as ribbon, colored blue to red from the N-terminus. The RNA primer and DNA template are shown in dark and light gray, respectively. The two deoxyguanosine nucleotides added to the RNA primer by Pol α during the crystallization experiment as shown in light gray. The dGTP nucleotide is shown in spacefill representation. The different subdomains of the polymerase structure are marked in the figure. (**B**) A view of the RNA primer/DNA template double helix bound to Pol α. The nucleic acids are shown as stick models; carbon atoms in light brown, nitrogen atoms in blue, oxygen atoms in red and phosphate atoms in orange. The 2'-hydroxyl oxygen atoms of the ribose moieties of the RNA primer are shown in magenta. The polymerase is shown as a clipped molecular surface, in blue. The conformation assigned to each dinucleotide step in the RNA/DNA helix is indicated on the left-hand side of the panel. The first five di-nucleotide steps of the hybrid RNA/DNA helix are numbered one to five, from the 3'-terminus of the RNA

*Figure 1. Continued on next page*

*Figure 1. Continued*

primer. (**C**) Scatter plot of $z_P$, the mean z-coordinate of the backbone phosphorus atoms with respect to individual dinucleotide reference frames, against the mean value for the four χ torsion angles at each di-nucleotide step, for the RNA/DNA helix bound to Pol α (black dots). The data points of the first five steps in the RNA/DNA helix bound to Pol α are numbered. For comparison, the values obtained for the DNA double helix bound to yeast Pol δ (empty dots; PDB entry 3IAY) and phage RB69 Pol (gray dots; PDB entry 1IG9) are also shown. For reference, the values obtained for A- and B-form DNA models based on fiber diffraction analysis are marked in the plot with a black cross. The stereochemical analysis of the RNA/DNA helix was performed with 3DNA (***Lu and Olson, 2003***).

The following figure supplements are available for figure 1:

**Figure supplement 1**. Primer extension assay for the D998N, R508A, N509A Pol α mutant.

**Figure supplement 2**. Mass spectrometry analysis of the RNA 10mer used in the crystallisation of the ternary complex.

**Figure supplement 3**. Details of the $2F_o$-$F_c$ electron density map at 3.1 Å resolution, after B-sharpening and contoured at 2.5 rmsd.

**Figure supplement 4**. Close-up view of Pol α's active site, showing the dGTP nucleotide and side chains of amino acids important for catalysis and nucleotide binding.

**Figure supplement 5**. Interaction of the single-stranded DNA template with Pol α.

**Figure supplement 6**. The actively copying Pol α crystallizes with two copies of the ternary complex in the asymmetric unit.

**Figure supplement 7**. Interaction with the non-crystallographic copy of Pol α captures the 5'-terminal nucleotide of the RNA primer in a surface pocket of the palm domain, in the crystals of ternary complex.

---

productively to RNA primer/DNA template shows that, although the palm domain makes a canonical set of interactions with the first 3 bp of the primer-template helix, the thumb domain engages almost exclusively with the RNA primer strand (***Figure 2A,B***).

The interactions between the thumb domain and the RNA primer are concentrated in a contiguous region of five nucleotides, from position two to six of the RNA primer and involve solely the ribose and phosphate moieties of the RNA backbone (***Figure 2C***). Two sequence motifs, residues 1074–1077 in the long segment of the L-shaped thumb, and residues 1130–1150 in the tip of the thumb, form a continuous protein interface that tracks the position of the phosphodiester backbone of the RNA primer. Pol α exploits the wide, shallow nature of the minor groove of the RNA/DNA helix with a series of hydrophobic contacts involving the side chains of Met1131, Leu1133, Tyr1140, Pro1141, Met1146, which recognize the C3'-endo conformation of the ribose moieties of RNA nucleotides in position four to six (***Figure 2D***).

The contiguous pair of invariant arginine residues 1075 and 1076 straddles the ribose-phosphate backbone between nucleotides three and four from the 3'-end of the RNA primer (***Figure 3***). The side chain of Arg1075 extends into the minor groove and packs its guanidinium moiety against the ribose sugar of Ade4 (***Figures 2C and 3A***), thus engaging in an interaction that matches closely the contact made by the equivalent Arg839 of yeast Pol δ with the DNA primer (***Figure 3B***). However, in the case of Pol α the close contact with Arg1075 induces a local rearrangement of the RNA conformation, which includes a B-DNA C2'-endo ribose pucker for Ade4 and unstacking between the aromatic bases of nucleotides four and five (***Figure 3A,C***). The local conformation of the RNA primer is stabilized by insertion of the Arg1076 side chain between the phosphate groups of nucleotides three and four (***Figures 2C and 3C,D***). The 'snap-clip' interaction of arginines 1075 and 1076 with the RNA backbone anchors the RNA primer to the thumb domain of Pol α, by facilitating the formation of three hydrogen bonds between the phosphate groups of nucleotides four, five and six and the mainchain nitrogens of residues Arg1076, Lys1132 and Ser1134, respectively (***Figure 3D***).

The hydrogen-bond network between thumb domain and RNA is augmented by polar interactions contributed by the side chains of Lys1132, Ser1134, Lys1135 and Tyr1140 (***Figure 3D***). The G1116E

**Table 1.** X-ray data processing and crystallographic refinement

### X-ray data processing

| | Seleno-methionine | | | Apo | | | Binary complex | | | Ternary complex | | |
|---|---|---|---|---|---|---|---|---|---|---|---|---|
| Space group | P 2 2$_1$ 2$_1$ | | | P2$_1$ | | | P1 | | | P 2$_1$ 2$_1$ 2$_1$ | | |
| Unit cell (Å) | 74.0 127.5 144.1 | | | 74.4 127.1 74.5 | | | 72.1 74.8 117.0 | | | 111.7 145.7 197.2 | | |
| Angles (°) | 90.0 90.0 90.0 | | | 90.0 104.8 90.0 | | | 82.3 72.6 82.4 | | | 90.0 90.0 90.0 | | |
| Mosaicity | 0.39 | | | 0.54 | | | 0.77 | | | 0.35 | | |
| | Overall | Innershell | Outershell | Overall | Innershell | Outershell | Overall | Innershell | Outershell | Overall | Innershell | Outershell |
| Low resolution limit (Å) | 73.97 | 73.97 | 2.81 | 53.51 | 53.51 | 2.42 | 58.77 | 58.77 | 3.16 | 49.39 | 49.39 | 3.26 |
| High resolution limit (Å) | 2.67 | 8.43 | 2.67 | 2.3 | 7.27 | 2.3 | 3 | 9.49 | 3 | 3.1 | 9.79 | 3.1 |
| $R_{merge}$ | 0.133 | 0.052 | 0.663 | 0.085 | 0.039 | 0.512 | 0.073 | 0.03 | 0.921 | 0.105 | 0.042 | 0.974 |
| $R_{merge}$ in top intensity bin | 0.061 | - | - | 0.051 | - | - | 0.029 | - | - | 0.046 | - | - |
| $R_{meas}$ (within I+/I−) | 0.14 | 0.054 | 0.703 | 0.104 | 0.048 | 0.621 | 0.085 | 0.035 | 1.068 | 0.113 | 0.046 | 1.05 |
| $R_{meas}$ (all I+ & I−) | 0.148 | 0.071 | 0.702 | 0.1 | 0.045 | 0.609 | 0.085 | 0.035 | 1.068 | 0.113 | 0.046 | 1.05 |
| $R_{pim}$ (within I+/I−) | 0.043 | 0.017 | 0.229 | 0.058 | 0.027 | 0.35 | 0.043 | 0.018 | 0.539 | 0.042 | 0.018 | 0.386 |
| $R_{pim}$ (all I+ & I−) | 0.033 | 0.016 | 0.165 | 0.04 | 0.019 | 0.246 | 0.043 | 0.018 | 0.539 | 0.042 | 0.018 | 0.386 |
| Fractional partial bias | −0.008 | −0.043 | −0.028 | −0.035 | −0.025 | −0.045 | −0.039 | −0.034 | −0.178 | −0.007 | −0.006 | −0.127 |
| Total number of observations | 781,855 | 25,826 | 97,175 | 355,357 | 10,427 | 50,930 | 175,884 | 5217 | 25,591 | 428,397 | 13,181 | 61,528 |
| Total number unique | 39,626 | 1401 | 5656 | 58,906 | 1764 | 8523 | 45,027 | 1413 | 6546 | 59,154 | 1990 | 8380 |
| Mean((I)/sd(I)) | 17.9 | 39.6 | 4.6 | 11.3 | 30.1 | 3 | 13.6 | 55.6 | 1.5 | 10.8 | 28.9 | 2.1 |
| Completeness | 100 | 99.6 | 99.8 | 99.3 | 91.4 | 99.1 | 97.8 | 96.7 | 97.5 | 99.7 | 97.2 | 98.4 |
| Multiplicity | 19.7 | 18.4 | 17.2 | 6 | 5.9 | 6 | 3.9 | 3.7 | 3.9 | 7.2 | 6.6 | 7.3 |
| Anomalous completeness | 99.9 | 99.7 | 99.5 | | | | | | | | | |
| Anomalous multiplicity | 10.3 | 11 | 8.8 | | | | | | | | | |
| DelAnom correlation between half-sets | 0.602 | 0.823 | 0.068 | | | | | | | | | |
| Mid-slope of Anom normal probability | 1.372 | - | - | | | | | | | | | |

### Crystallographic refinement

| | Seleno-methionine | Apo | Binary complex | Ternary complex |
|---|---|---|---|---|
| R-factor (overall/outershell) | 0.1959 (0.2405) | 0.2051 (0.3079) | 0.2551 (0.3623) | 0.2111 (0.3570) |
| R-free (overall/outershell) | 0.2341 (0.3058) | 0.2361 (0.3453) | 0.2858 (0.4040) | 0.2479 (0.3909) |
| Number of atoms | 13,497 | 13,690 | 27,707 | 29,108 |
| macromolecules | 6637 | 6683 | 13,914 | 14,680 |
| ligands | | | | 62 |
| water | 110 | 216 | 0 | 0 |
| Protein residues | 827 | 829 | 1687 | 1754 |
| RMS(bonds) | 0.002 | 0.002 | 0.003 | 0.003 |
| RMS(angles) | 0.57 | 0.59 | 0.77 | 0.75 |
| Ramachandran favored (%) | 97 | 97 | 96 | 94 |

*Table 1. Continued on next page*

*Table 1. Continued*

| | | | | |
|---|---|---|---|---|
| Ramachandran outliers (%) | 0.12 | 0 | 0.061 | 0.24 |
| Clashscore | 3.74 | 2.75 | 10.04 | 11.55 |
| Wilson B-factor | 45.94 | 42.31 | 91.64 | 97.34 |
| Average B-factor | 56.5 | 62.4 | 108 | 111.6 |
| macromolecules | 56.7 | 62.7 | 108 | 111.6 |
| solvent | 45.8 | 54.9 | | |

mutation in fission yeast Pol α at the equivalent position to Ser1134 causes a defect in mating-type switching (*Singh and Klar, 1993*): the close interaction of Ser1134 with the RNA primer suggests that the defect might be caused by weakened affinity of Pol α for its primer-template substrate.

## Limited extension of the RNA primer

The crystallographic analysis pointed to a specific mechanism for the intrinsic and induced recognition of the templated RNA primer by Pol α. The result of the structural study prompted us to investigate whether the polymerase activity of Pol α reflected such specificity. In order to test this hypothesis, we examined the ability of the polymerase domain of yeast Pol α to extend with deoxynucleotides a templated primer made of either RNA or DNA. Whereas Pol α displayed only weak activity and limited processivity when extending a DNA primer, extension of an RNA primer resulted in much higher levels of polymerization (*Figure 4A*). Strikingly, the profile of RNA-dependent extension products showed a pronounced peak at between 10 to 12 nucleotides, indicating strong processivity limited to a full-turn of double helix (*Figure 4A,B*). Such characteristics of the catalytic activity of Pol α were observed for the isolated polymerase core as well as the Pol α /primase complex and are independent of primer size (*Figure 4C*).

The marked difference in catalytic behavior of Pol α in the presence of either an RNA or a DNA primer is consistent with the crystallographic evidence that Pol α recognizes structural features of the primer-template helix. Elongation of the RNA primer with deoxynucleotides would cause translocation of the polymerase beyond the RNA/DNA duplex region and onto B-DNA, with loss of the RNA-specific contacts, eventually causing Pol α to stall and terminate primer synthesis. We sought to test this mechanism of primer termination by design of DNA primer sequences with altered conformational properties. It is known that, although double-stranded DNA normally adopts a B-DNA conformation, its structure can display pronounced conformational variation depending on the local sequence of bases. Short runs of consecutive deoxyguanosine nucleotides can induce A-like DNA conformation in double-stranded DNA (*Svozil et al., 2008*; *Marathe et al., 2009*). We examined the ability of Pol α to extend a templated oligo(dA) primer with increasing deoxyguanylate content, introduced incrementally as di- and tri-nucleotides. Indeed, as the deoxyguanosine content of the DNA primer increased, the size profile of the extension products synthesized by Pol α changed from that expected of a DNA primer to that of an RNA primer (*Figure 4D*).

## RNA-DNA primer termination and release

The observation that extension of the RNA primer is limited to a single turn of double helix indicates that termination of primer synthesis might be triggered by the loss of specific interactions of Pol α with the RNA/DNA duplex. Further insight as to the possible structural basis for a termination mechanism can be obtained by comparing the structures of isolated and actively copying Pol α, which reveals a striking difference in the position of the thumb domain. In order to adopt the conformation observed in ternary complex, the thumb domain of Pol α must rotate inwards by about 20°, a large conformational rearrangement compared for instance with the limited movement of the thumb domain observed in structures of isolated and copying RB69 Pol (*Figure 5A*). The large difference in the position of the thumb domain in the ternary complex and in the isolated polymerase suggests a molecular basis for the release of the completed primer-template by Pol α: loss of the interactions with the primer-template duplex would allow the thumb domain to rotate away from the DNA, terminating the productive engagement of the polymerase with its substrate.

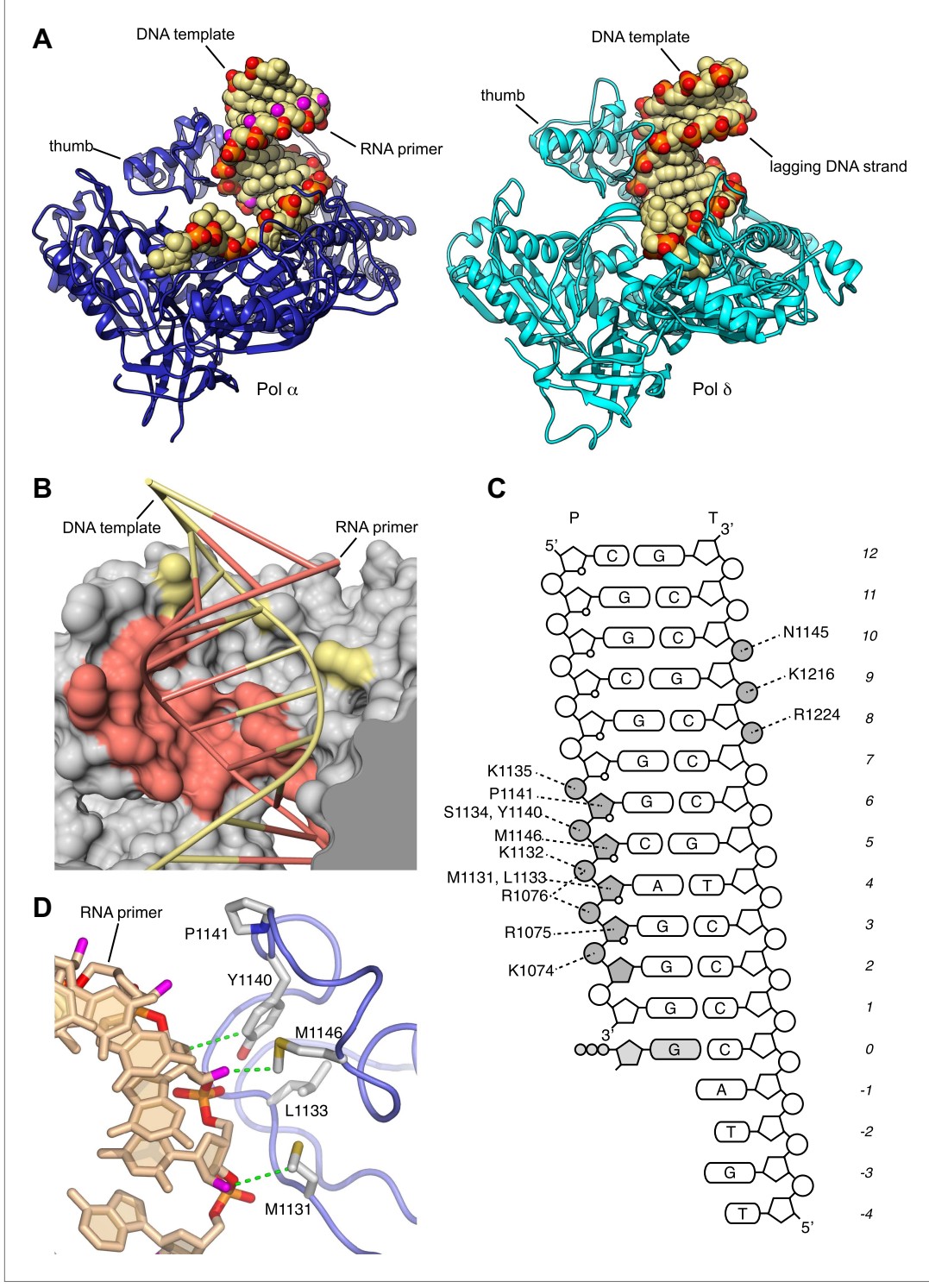

**Figure 2**. Specific recognition of the RNA/DNA helix by Pol α. (**A**) Identical views of active complexes of yeast Pols α (left) and δ (*Swan et al., 2009*; right; PDB entry 3IAY), bound to primer/template duplexes and incoming deoxynucleotide. The polymerase structures are depicted as ribbons (Pol α, blue; Pol δ, cyan). Primer/template duplexes are shown in spacefill representation and colored light brown, with the exception of the phosphate groups that have red oxygen atoms and orange phosphate atoms. The 2′-hydroxyl oxygen atoms in the ribose moieties of the RNA are highlighted in magenta. (**B**) The interface between the thumb domain of Pol α and the RNA primer/DNA template duplex. The ribo-phosphate backbone is shown as a thin tube, in salmon and pale
*Figure 2. Continued on next page*

*Figure 2. Continued*

yellow color for the RNA and DNA strands, respectively, and the bases are depicted as rungs of a ladder. Pol α is shown as a molecular surface, colored gray except for atoms that are within a 5 Å radius of the RNA primer or the DNA template, that are colored as the nucleic acid strand. (**C**) Schematic diagram of the interactions of Pol α's thumb domain with the RNA/DNA helix. The ribophosphate portion of nucleotides that interact with the thumb domain is shaded gray. A small circle at position two of the ribose ring indicates the RNA nucleotides. (**D**) Hydrophobic contacts of the thumb domain with the exposed minor groove of the RNA/DNA helix. The protein is shown as thin blue tube and amino acid side chains as sticks with white carbon atoms, yellow sulfur atoms and red oxygen atoms. The RNA primer is shown as sticks, with atoms colored as in panel A. Protein-RNA contacts are shown as dashed green lines.

We explored this model by performing nanosecond (ns) molecular dynamics simulations of the catalytic domain of Pol α, based on the crystallographic models of isolated and actively copying polymerase (*Figure 5B* and *Figure 5—figure supplement 1*). Principal component analysis of the trajectories shows that the actively copying polymerase maintains a stable conformation over the 100 ns duration of the simulation. In comparison, the structure of the isolated polymerase displays considerable conformational fluctuations, indicative of large-scale conformational changes and local loss of secondary structure that affect in particular the N-amino terminal region and the thumb domain. Importantly, the conformational flexibility of the isolated polymerase does not seem to be sufficient to sample conformations adopted by the polymerase in the ternary complex.

In order to determine whether the conformation of the actively copying polymerase can be maintained in the absence of the RNA/DNA duplex and deoxynucleotide, we analyzed the behavior of the polymerase taking the conformation adopted in the ternary complex as the starting point for the simulation. The trajectory showed an increased conformational instability, accompanied by a tendency to relax towards the apo conformation of the polymerase (*Figure 5C* and *Video 1*). Taken together, the results of the molecular dynamics simulations indicate that Pol α reaches the active conformation observed in the ternary complex via an 'induced fit' mechanism that is dependent on its interactions with the RNA/DNA duplex and deoxynucleotide. Loss of the optimal set of protein-RNA contacts would weaken the grip of the polymerase on its primer/template substrate and promote primer release.

## Palm domain movement controls nucleotide access to active site

It is well established that adoption of a conformation competent for catalysis by DNA polymerases requires the concerted movements of the fingers domain, that rotates upwards to form the nucleotide binding site, and of the thumb domain, that embraces the primer-template duplex. Comparison of Pol α structures in the apo, binary and ternary complex reveals a further and unexpected rearrangement: as the polymerase morphs from the apo to the active conformation, the palm domain tilts away from the rising fingers, by rotating on the short helix containing the highly conserved 864-DFNSLYPS-871 motif, known as region II of B-family DNA polymerases (*Wang et al., 1989*; *Delarue et al., 1990*; *Hubscher et al., 2002*; *Figure 6A*, *Figure 6—figure supplement 1*, *Video 2*).

Rotation of the palm domain is accompanied by a conformational rearrangement of region II that is necessary in order to accommodate the phosphoribose moiety of the incoming nucleotide (*Figure 6B*). The two copies of binary complex present in the asymmetric unit of the crystals capture intermediate states of the transition, as in one binary complex the conformation of region II is apo-like, whereas in the second copy region II adopts a conformation akin to that observed in the ternary complex. The range of positions adopted by the palm domain in the apo, binary and ternary complex structures suggests that adoption of an active conformation by Pol α might require a wider set of conformational rearrangements than predicted by current models of DNA polymerase activity. The observed movement of the palm domain could participate in a 'ratchet' mechanism for unidirectional translocation of Pol α, as return of the palm domain to the apo conformation after catalysis would propel the polymerase forward onto the next templating base. Interestingly, site-specific mutations of Ser863 in region II of human Pol α (Ser867 of yeast Pol α), which acts as the pivot during rotation of the palm domain, cause a drastic reduction in the specific activity of the polymerase without changing its kinetic parameters, in agreement with a potential structural role of this serine residue in DNA polymerisation (*Dong et al., 1993*).

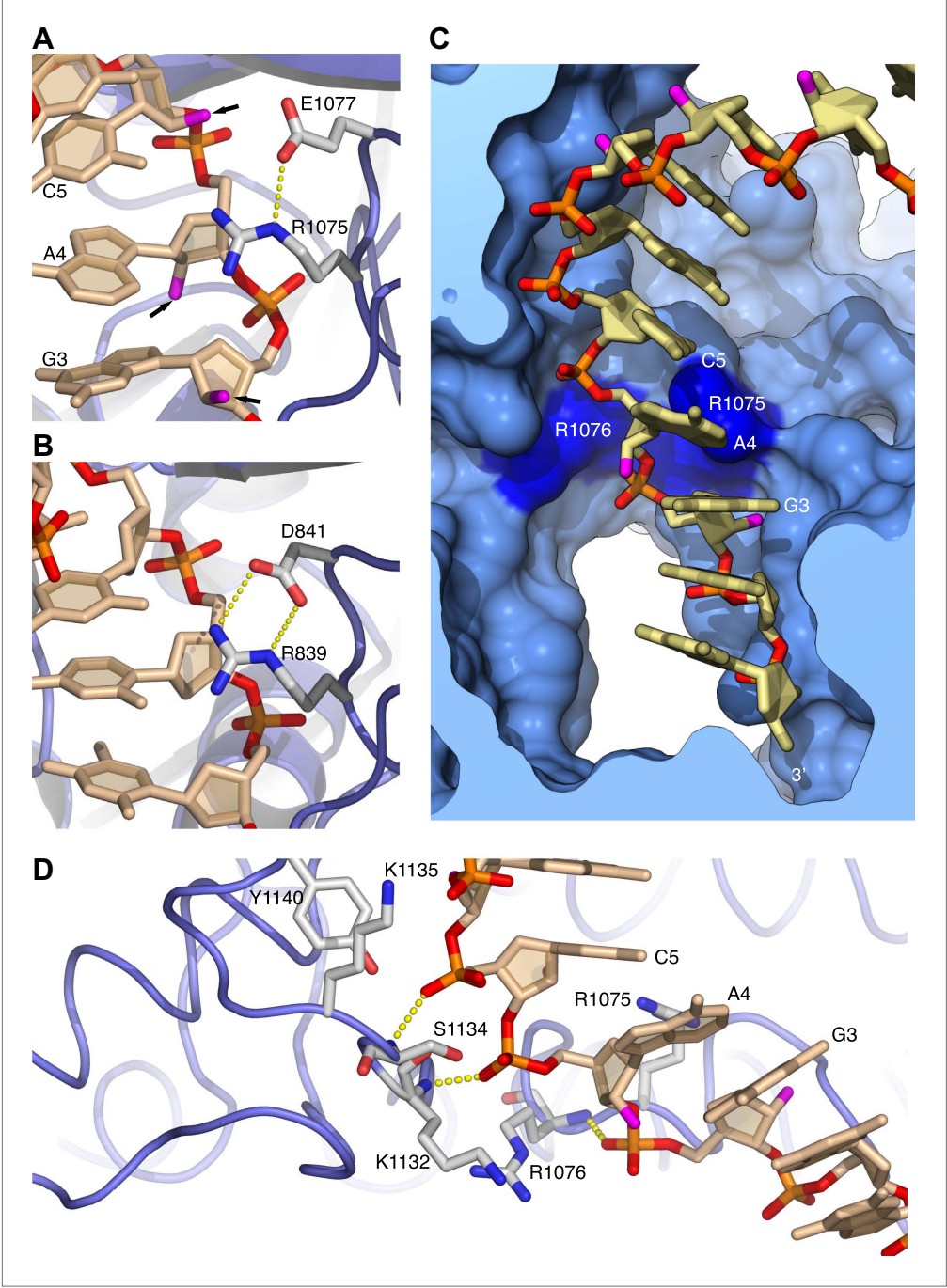

**Figure 3**. Arginines 1075 and 1076 anchor Pol α to the RNA primer. (**A**) Close-up view of the interaction of Arg1075 with the RNA primer. A hydrogen bond interaction between the carboxylate group of Glu1077 and Arg1075 is also shown, as a yellow dashed line. The protein is shown as thin blue tube and amino acid side chains as sticks with white carbon atoms, blue nitrogen atoms and red oxygen atoms. The RNA primer is shown as sticks, with atoms colored as in *Figure 2A*. A black arrow highlights the position of the 2'-hydroxyl moiety of the ribose ring. For clarity, the DNA template has been omitted. (**B**) Close-up view of the interaction of Pol δ Arg839 with the DNA primer, in the same orientation as Pol α in panel A. The interaction of Asp841 with Arg839 is also shown. (**C**) Conformation of the RNA primer bound to Pol α. The polymerase is shown as a clipped molecular surface, in light blue. The position of Arg1075 and Arg1076 is indicated in dark blue. Color scheme is as in *Figure 2A*. The DNA template strand is omitted for clarity. (**D**) Polar contacts at the interface between Pol α's thumb domain and the RNA primer. Hydrogen bonds between the phosphate groups of the RNA primer and main chain nitrogen atoms of the thumb domain are depicted as dashed yellow lines. Color scheme is as in *Figure 2A*. The DNA template strand is omitted for clarity.

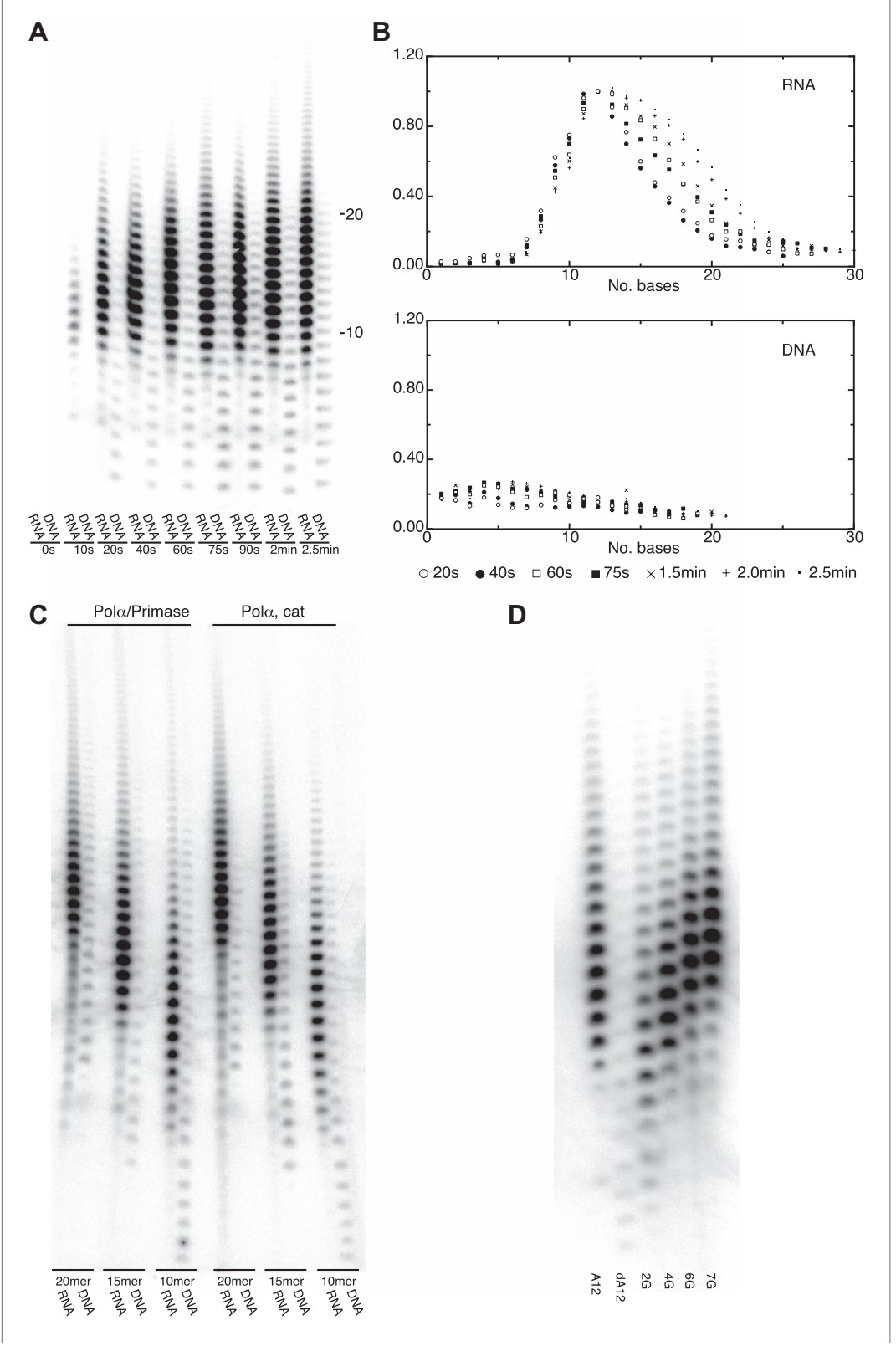

**Figure 4**. DNA- and RNA-primed dNTP polymerization by Pol α. (**A**) Primer extension assay, analyzed by denaturing acrylamide gel electrophoresis and [α-³²P]dATP phosphorimaging. The wild-type polymerase domain of yeast Pol α was incubated at 37°C with poly(dT) 70mer template, dATP and either an oligo(A) or oligo(dA) 15mer primer
*Figure 4. Continued on next page*

*Figure 4. Continued*

(see 'Materials and methods' for details). The reaction products were analyzed at the indicated time points. (**B**) Quantitative product size-distribution analysis of the primer extension assay in panel A. For each time point, the intensities of the RNA and DNA primer extension products were measured and normalized relative to the measured intensity of the 12mer product in the RNA primer lane. The normalized values of the RNA and DNA-primer extension products are plotted against the size of the extension product, as number of bases, in the top and bottom panels, respectively. The normalized values are the average of three independent measurements. (**C**) Primer extension assay. The catalytic domain of yeast Pol α and a recombinant version of the yeast Pol α/primase complex were incubated at 37°C and for 60 s with a poly(dT) 70mer template, dATP and either an oligo(A) or oligo(dA) of different length, as indicated in the panel. (**D**) Primer extension assay. The catalytic domain of yeast Pol α was incubated at 37°C and for 60 s with dATP and one of the following 5′-to-3′ primer/template pairs: A12, $A_{12}$/(dT)$_{50}$; dA12, (dA)$_{12}$/(dT)$_{50}$; 2 G, $A_5GGA_5$/$T_{43}CCT_5$; 4 G, $AGGA_2GGA_5$/$T_{43}CCT_2CCT$ ; 6 G, $AGGA_2GGAGGA_2$/$T_{40}CCTCCT_2CCT$; 7 G, $AGGAGGGAGGA_2$/$T_{40}CCTCCCTCCT$.

## Discussion

Here we have presented structural, biochemical and computational experiments that address the essential role of Pol α in the initiation of DNA synthesis in eukaryotic replication. Our findings provide evidence for a specific mechanism of RNA primer recognition, limited extension and termination by Pol α, that is encoded in the interaction of the polymerase with the hybrid RNA primer/DNA template helix (*Figure 7*). Recognition of the intrinsic and induced geometry of the RNA/DNA helix by Pol α would prompt rapid dNTP polymerization, until a full turn of DNA double helix has been synthesized. Translocation away from the RNA/DNA helix and synthesis of B-form DNA would cause Pol α to stall, akin to a locomotive running into railtrack of different gauge. Loss of the optimal interface contacts between Pol α and the primer-template helix would favor a conformational change towards the apo conformation of the polymerase, with consequent disengagement of Pol α from the completed RNA-DNA primer.

The emerging picture of Pol α as a specialized polymerase that extends processively an RNA primer with deoxynucleotides to the limited size of a helical turn has important implications for our understanding of lagging strand synthesis. Uninterrupted DNA synthesis during replication depends on the tight coordination of Pol α function with the upstream activity of primase, that synthesizes the RNA primer, and the downstream activity of Pol δ, that elongates the resulting RNA-DNA primers. The efficient mechanism for the rapid extension and controlled termination of RNA primers by Pol α uncovered here seems ideally suited for the frequently repeated, highly integrated priming events taking place on the lagging strand template. After termination of synthesis and release of the completed primer, the physical tethering of Pol α to primase would facilitate its recycling onto the 3′-end of the next RNA primer (*Nunez-Ramirez et al., 2011*; *Kilkenny et al., 2012*).

The evidence for a mechanism of primer release which stems from Pol α's ability to discriminate between different shapes of the primer-template helix is relevant to our mechanistic understanding of the coordinated sequence of events that take place during Okazaki fragment synthesis. Spontaneous primer release once a helical equivalent of deoxynucleotides has been added to the RNA primer would make the 3′-terminus of the RNA-DNA primer directly available for extension by Pol δ and Pol ε, the DNA polymerases that synthesize the bulk of genomic DNA. Such a mechanism provides a straightforward model of primer hand-off in eukaryotic replication, by removing the need for molecular transactions between primer-bound Pol α and the replicative apparatus deputed to leading and lagging strand synthesis. We note that, although the molecular details differ, the mechanism of primer release by Pol α described here is conceptually similar to the mechanism of polymerase switching proposed for Pol η after lesion by-pass (*Biertumpfel et al., 2010*; *Silverstein et al., 2010*): for both polymerases, termination of DNA synthesis appears to be a direct consequence of the specific way in which they interact with their nucleic acid substrate.

The ability of primases to synthesise RNA primers of discrete size has been well characterized (*Kuchta and Stengel, 2010*). Our work has highlighted a previously unappreciated functional similarity between primase and Pol α, as extension of an RNA primer by Pol α leads to controlled termination of synthesis, in a functionally analogous manner to the known counting ability of primase. This functional symmetry in the behavior of the two polymerases responsible for priming DNA synthesis seems

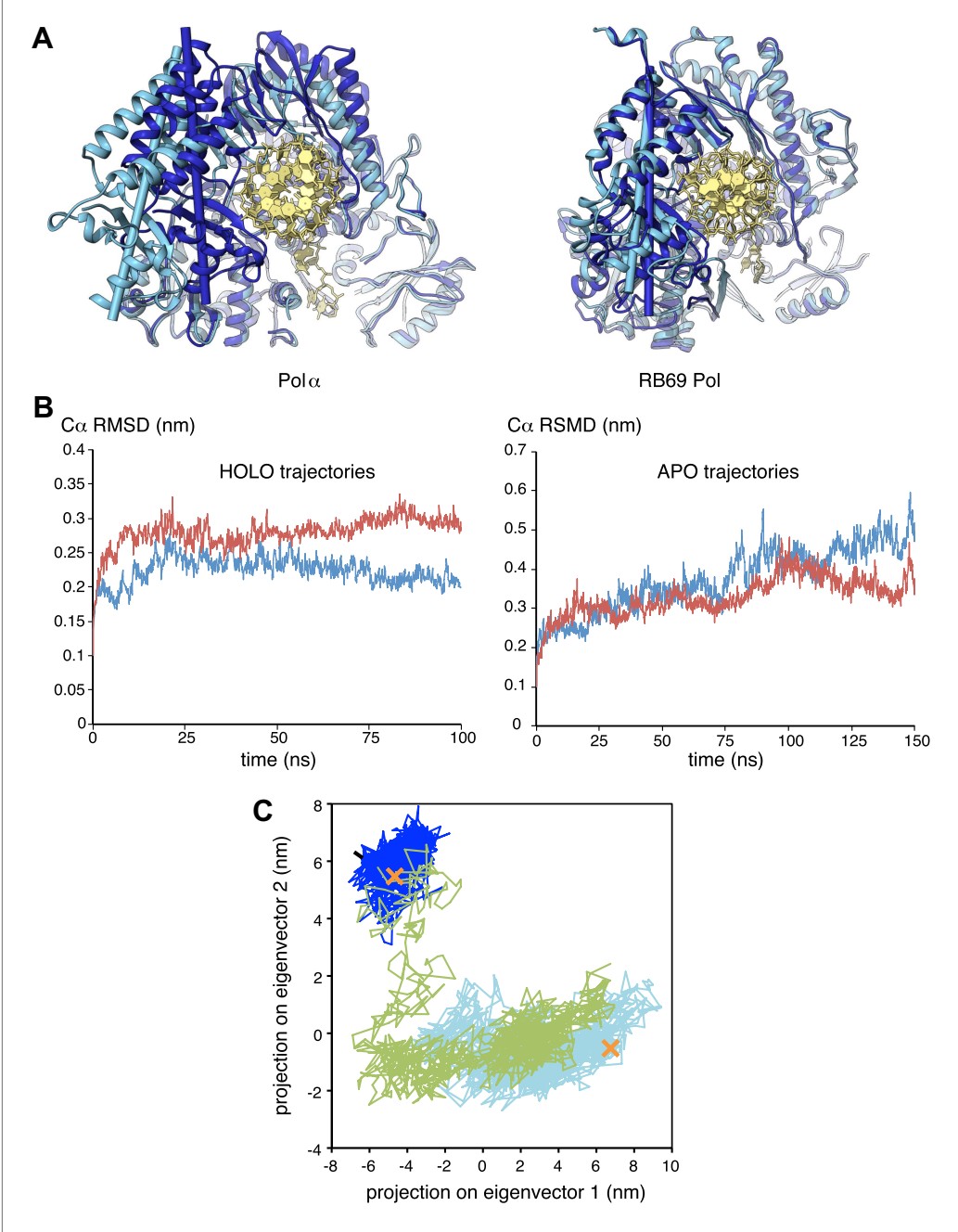

**Figure 5**. Conformational changes of Pol α during DNA priming. (**A**) Structural superposition of the isolated and active conformations of the polymerase domain of Pol α (left) and RB69 Pol (***Wang et al., 1997***; ***Franklin et al., 2001***; right; PDB entries 1IH7 and 1IG9). The protein is shown as ribbons and the nucleic acid duplex as sticks. The polymerase domain is colored light blue (apo) and dark blue (active conformation). In order to help visualize the difference between apo and active conformations of the polymerase, the major inertia axis of the thumb domain for each structure is also shown (***Pettersen et al., 2004***). (**B**) Change in root mean square deviation (RMSD) of the Cα positions over the time of simulation for the HOLO (100 ns; left panel) and APO trajectories (150 ns; right panel), calculated relative to the starting structure of the trajectory (see 'Materials and methods' for details). (**C**) Time-dependent projection on the first two principal components of the trajectory of the apo polymerase structure (150 ns, light blue trace), the polymerase structure in the ternary complex (100 ns, dark blue trace) and the polymerase structure starting from the conformation adopted in the ternary complex, but in absence of the RNA/DNA duplex and deoxynucleotide (100 ns; green trace). A black arrow marks the starting position of the

*Figure 5. Continued on next page*

*Figure 5. Continued*
green trace. The position of the crystallographic models of the polymerase in the apo form and in the ternary complex is marked by an orange cross.
The following figure supplements are available for figure 5:

**Figure supplement 1**. Difference between the root mean square fluctuation (RMSF) values of the APO and HOLO trajectories at each Cα position of the polymerase structure.

logical, as the lack of proof-reading ability by Pol α poses a potential threat to genomic stability. Thus, in addition to excising the RNA portion of the primer, eukaryotic replication faces the additional challenge of correcting possible mismatches introduced by Pol α in the DNA segment of the primer. How mistakes made by Pol α are corrected by the replication apparatus has not been fully elucidated yet but current evidence points to a proofreading role of Pol δ (*Pavlov et al., 2006*). The mechanism of primer termination indicated by our findings would constitute a simple and elegant way to limit the extent of inaccurate DNA that needs to be corrected from the 5′-terminus of each Okazaki fragment.

## Materials and methods

### Cloning, expression and purification

A gene segment corresponding to amino acids 349–1258 of the yeast Pol α (910 residues; hereinafter referred to as Pol α) was PCR amplified from *Saccharomyces cerevisiae* genomic DNA and inserted by enzymatic restriction into the pRSFDuet-1 vector (Merck KGaA, Darmstadt, Germany) for bacterial over-expression, as a polypeptide fused at its N-terminus to a dual histidine and streptavidin TEV-cleavable tag. The pRSF-Pol α construct was over-expressed with IPTG in *Escherichia coli* BL21(DE3)Rosetta2 strain (Invitrogen Life Technologies Ltd, Paisley, UK) growing at 20°C in Turbo Broth™ (Molecular Dimensions Ltd, Newmarket, UK). The Pol α protein was purified by successive steps of Co-NTA affinity chromatography (Qiagen Ltd, Manchester, UK), Heparin Sepharose chromatography (GE Healthcare Life Sciences, Little Chalfont, UK) and Strep-Tactin chromatography (IBA GmbH, Göttingen, Germany), followed by tag removal and a final step of size exclusion chromatography on a Superdex 200 16/60 (GE Healthcare). The eluted Pol α sample was concentrated to 100 μM in 20 mM Hepes pH 6.8, 300 mM NaCl, 10% glycerol buffer and divided in small aliquots that were flash frozen in liquid nitrogen for crystallography and functional studies.

A version of pRSFDuet-1-Pol α carrying mutations R508A, N509A, D998N construct was prepared with QuikChange site-directed mutagenesis kit (Agilent Technologies UK Ltd, Stockport, UK), according to the manufacturer's instructions. Catalytic Asp998 was mutated to asparagine in order to produce an inactive polymerase; unexpectedly, the D998N mutation greatly reduced but did not abolished polymerisation by Pol α (*Figure 1—figure supplement 1*). The double mutation R508A, N509A was introduced in order to promote crystallization of the ternary complex. As expression levels of the Pol α mutant were lower than observed for the wild-type protein, over-expression was carried out in a 30 l BIOSTAT® (Sartorius Stedim UK Ltd, Epsom, UK) bioreactor. Purification of the Pol α mutant was carried out in the same way as for the wild-type protein.

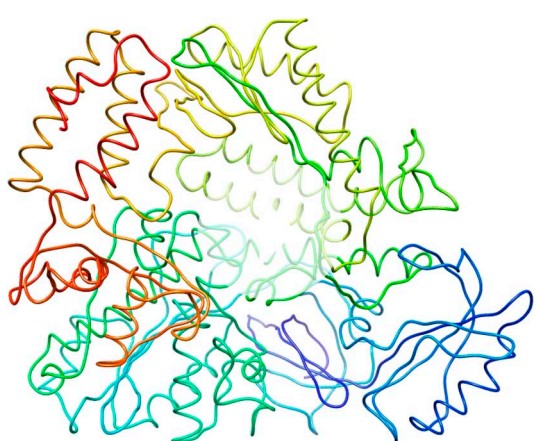

**Video 1**. Principal motion of the Pol α structure in the simulated trajectory of the polymerase, starting from the conformation adopted in the ternary complex but in the absence of RNA/DNA and dGTP (APOFORCED trajectory; see 'Materials and methods' for details). The movie was prepared with the MD movie tool in Chimera, by interpolation of 30 Pol α structures representative of the APOFORCED trajectory. The Pol α structure is shown as a ribbon, coloured blue to red from the N- to the C-terminal end of the polypeptide.

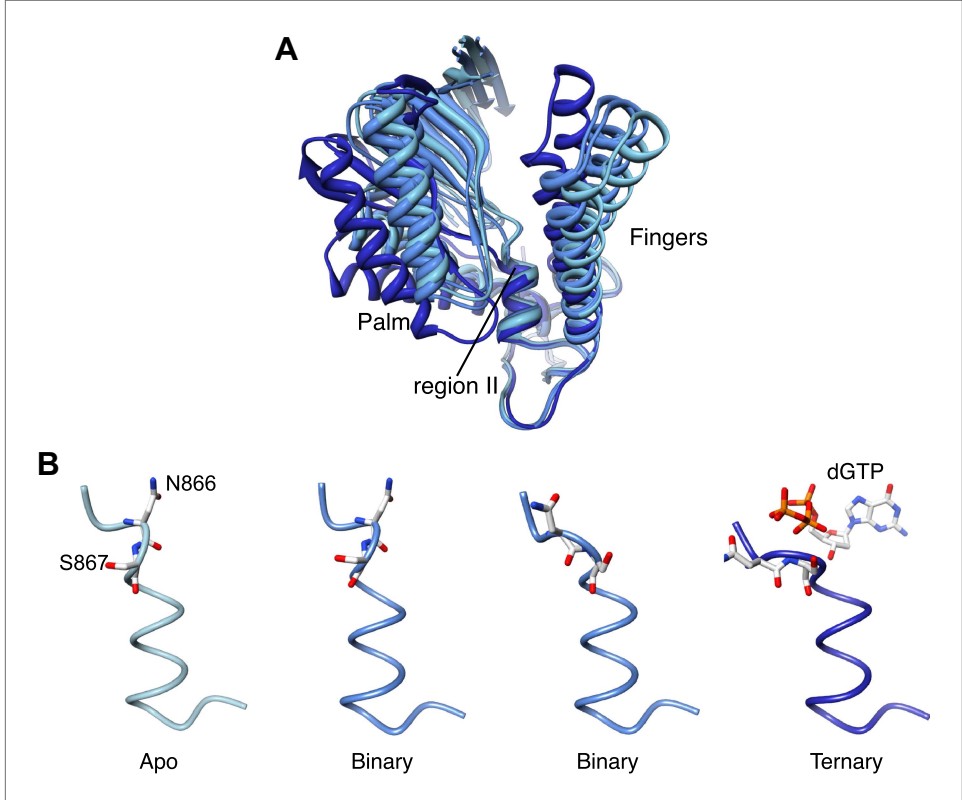

**Figure 6**. Palm domain movement controls nucleotide access to active site. (**A**) Structural superposition of the polymerase domain of Pol α in the apo form (light blue), in the binary (blue) and ternary (dark blue) complex. Only the palm and fingers domains of the polymerase are shown. The position of the region II sequence, conserved in B-family DNA polymerases, is indicated. (**B**) Polypeptide conformation in the region II sequence of Pol α. From left to right, the panel shows the conformation of the polymerase in the apo form (light blue), in the two polymerase molecules in the asymmetric unit of the binary complex crystals (blue) and in the ternary complex form (dark blue). The polypeptide is shown as a thin tube. The 866-NS-867 di-peptide and the dGTP nucleotide are shown as sticks.

The following figure supplements are available for figure 6:

**Figure supplement 1**. Structures of Pol α in the apo form, in a binary complex with RNA/DNA and in a ternary complex with RNA/DNA and dGTP.

For preparation of recombinant Pol α/primase complex, the pRSFDuet-1 vector was adapted for polycistronic expression of Pol α, the B subunit and the heterodimeric primase (RP and LP, unpublished data). Expression of the Pol α/primase complex was carried out in 30 l BIOSTAT® (Sartorius Stedim) bioreactor, using 20 l of auto induction media. Purification of the Pol α/primase complex was carried out according to a similar protocol followed for Pol α purification, including steps of Ni-NTA chromatography (Qiagen), Heparin sepharose chromatography (GE Healthcare), Strep-Tactin chromatography (IBA), tag removal by TEV cleavage and gel filtration chromatography. All steps of this purification were carried out at 4°C to avoid degradation of the complex. The purified Pol α/primase complex was concentrated to 20 μM in 20 mM Hepes pH 7.0, 300 mM KCl, 10% glycerol buffer and divided in small aliquots that were flash frozen in liquid nitrogen for functional studies.

## Crystal structure of Pol α

Crystals of Pol α (apo) were grown by mixing equal volumes of protein at 50 μM and 0.1 M Bicine pH 9.4, 6–10% PEG 8000 and improved by streak seeding. The X-ray crystal structure of Pol α was determined by single-wavelength anomalous scattering using selenomethionine-labelled (SeMet) protein crystals. Pol α crystallized in the monoclinic space group $P2_1$, with cell dimensions: a = 74.4 Å b = 127.1 Å c = 74.5 Å, β = 104.8°. X-ray diffraction data for native and SeMet crystals were collected

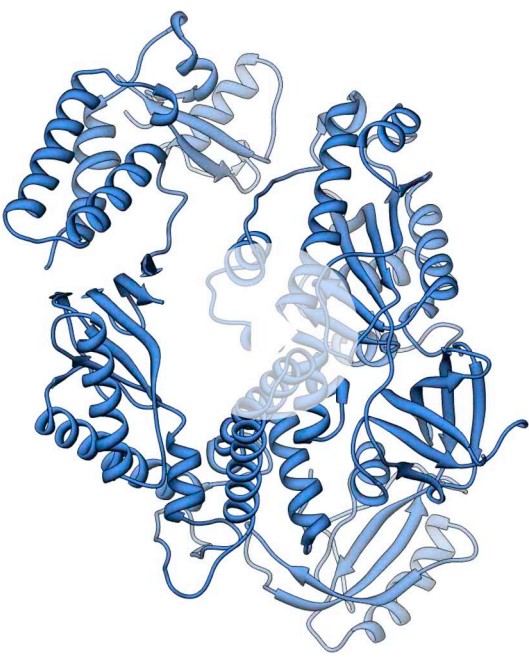

**Video 2**. Animation that shows morphing between the Pol α structures in the ternary complex and in the apo form. Pol α is shown in ribbon representation, coloured light blue. The animation was prepared with the MD movie tool in Chimera.

at European Synchrotron Research Facility (ESRF) on beam line ID23-1 at the peak wavelength of the Selenium K edge. Data were indexed, integrated, scaled and merged using MOSFLM (*Battye et al., 2011*) and SCALA (*Evans, 2011*) of the CCP4 program suite (*Collaborative Computational Project, 1994*). Phasing and initial automatic model building of SeMet Pol α was carried out in PHENIX (*Adams et al., 2010*) and the crystallographic model was completed manually and refined at 2.67 Å resolution in REFMAC 5.5 (*Murshudov et al., 1997*), BUSTER (*Bricogne et al., 2011*) and PHENIX, to R-factor/R-free (%) of 19.6/23.4. As SeMet Pol α crystallized in a different space group (orthorhombic space group $P22_12_1$, cell dimensions: 74.0 Å 127.4 Å 144.1 Å) determination of the native Pol α structure was achieved by molecular replacement with PHASER (*McCoy et al., 2007*), using the crystal structure of SeMet Pol α as search model. The crystal structure of native Pol α structure was refined at 2.3 Å resolution using BUSTER and PHENIX, to R-factor/R-free (%) of 20.5/23.6. For both SeMet and native Pol α structures, amino acids 677–679 (3 residues), 816–847 (32 residues), 1056–1062 (7 residues), 1176–1185 (10 residues) and 1129–1258 (30 residues) were not visible or poorly ordered in the electron density maps during refinement and were therefore excluded from the final refined model. Details of data processing and crystallographic refinement are provided in *Table 1*.

## Co-crystal structure of Pol α with RNA/DNA

Co-crystals of Pol α bound to an RNA/DNA duplex (binary complex) were grown by vapour diffusion in hanging drop at 18°C, by mixing Pol α with 5′-AGGCGGGCAG-3′ RNA 10mer annealed to 5′-TTTTCGCTGCCCGCCT-3′ DNA 16mer and 1 mM di-deoxycytidine triphosphate with 0.1 M Bicine pH 8.0, 12% PEG 3350 and 10 mM MgCl₂. The binary complex crystallized in the triclinic space group P1, with cell dimensions: a = 72.1 Å b = 74.8 Å c = 117.0 Å α = 82.3° β = 72.6° γ = 82.4°, and two copies of the binary complex in the asymmetric unit. X-ray diffraction data were collected at ESRF beam line ID14-1, and processed as for the native Pol α crystal structure. The structure was solved by molecular replacement in PHASER, using the crystallographic model of the apo Pol α as search model, and refined to 3.0 Å resolution with PHENIX, to R-factor/R-free (%) of 25.5/28.6. Amino acids 349–350 (2 residues), 816–847 (32 residues), 1176–1186 (11 residues), 1243–1258 (16 residues) in both polymerase chains were not visible or poorly ordered in the electron density maps during refinement and were therefore excluded from the final refined model. In addition, amino acids 1056–1062 (7 residues) and 1133–1166 (34 residues) in polymerase chain B were disordered and excluded from the final model. Of the DNA 16mer/RNA 10mer substrate, only a duplex region spanning 8 bp from the 3′-terminus of the RNA primer was visible in the electron density map and was included in the final model. Three RNA nucleotides at the 5′-end of chains D, F and three DNA nucleotides at the 5′-end of chain E are poorly ordered in the map and their position must be considered tentative. Despite the presence of nucleotide in the crystallization buffer, no electron density for ddCTP was visible in the map. Details of data processing and the crystallographic refinement are provided in *Table 1*.

## Co-crystal structure of Pol α with RNA/DNA and dGTP

For co-crystallization of Pol α with RNA/DNA and deoxynucleotide (ternary complex), the R508A, N509A, D998N Pol α mutant was used. The ternary complex was reconstituted prior to crystallization

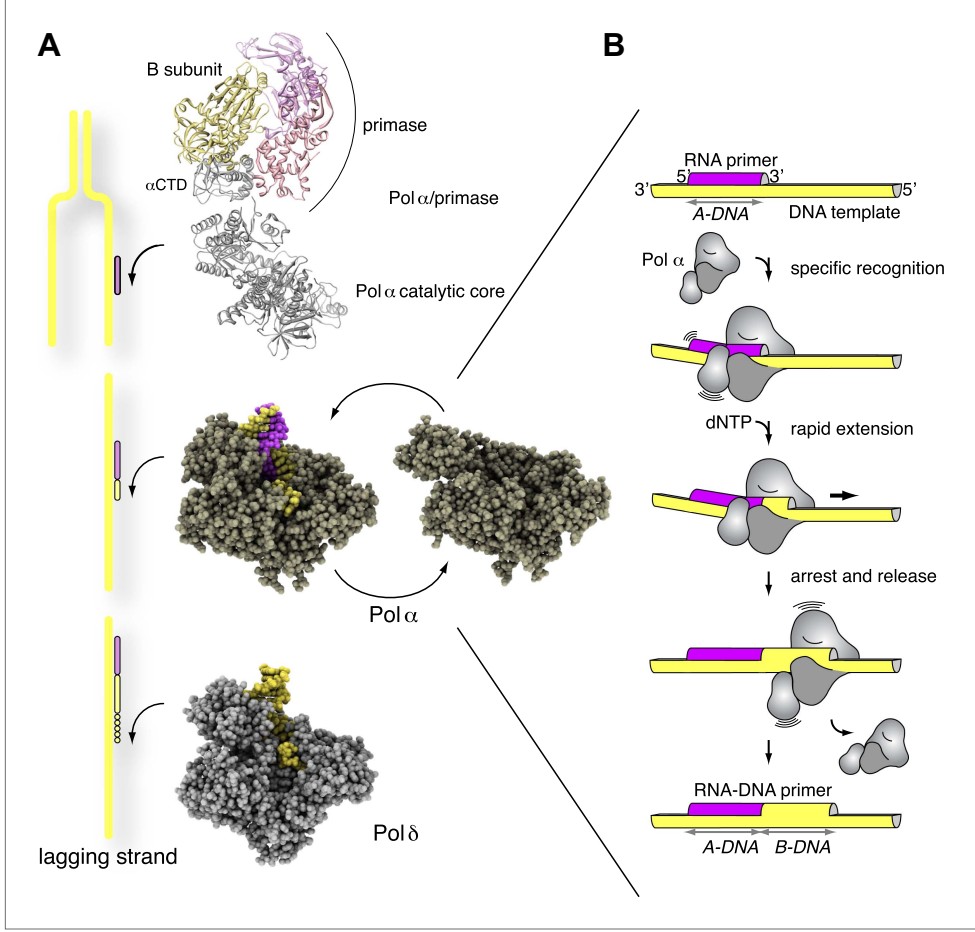

**Figure 7**. Mechanism of DNA primer synthesis by Pol α. (**A**) Priming eukaryotic replication requires synthesis of an RNA primer by the primase subunit of the Pol α/primase complex, intramolecular primer hand-off to the catalytic subunit of Pol α and limited primer extension with DNA; processive elongation of the completed RNA-DNA primer by Pol δ results in Okazaki fragment synthesis. The ribbon model of the Pol α/primase complex was assembled from the crystal structures of apo Pol α (this manuscript), the yeast Pol α CTD–B subunit complex (***Klinge et al., 2009***), and the archaeal primase (***Lao-Sirieix et al., 2005a***). The location of the Pol α/primase complex components in the model is based on previous electron microscopy and biochemical data (***Nunez-Ramirez et al., 2011***; ***Kilkenny et al., 2012***) and is not meant to give an accurate representation of their reciprocal spatial relationship. The crystal structures of isolated and copying Pol α and Pol δ are shown in spacefill representation. The model of Pol δ is based on PDB entry 3IAY (***Swan et al., 2009***). In the diagram, DNA is shown in yellow and RNA in magenta. (**B**) The catalytic cycle of Pol α consists of specific recognition of the templated RNA primer, rapid and limited primer extension with dNTPs and termination of synthesis induced by polymerization of B-form DNA. Constitutive tethering of primase to Pol α (not drawn in the panel; ***Kilkenny et al., 2012***) ensures efficient recycling of Pol α onto the 3'-terminus of the next RNA primer.

by incubating Pol α with a twofold excess of 5'-CGGCGGGCAG-3' RNA 10mer annealed to 5'-TGAGCGTGTGTACCCCTGCCCGCCG-3' DNA 25mer and 5 mM deoxyguanosine triphosphate. Crystals of ternary complex were grown under oil in micro-batch setup, by mixing the sample with 200 mM MgAc$_2$, 10% PEG 8000 crystallization buffer, at a protein to buffer ratio of 1.1:1.8 (v/v). Crystals were optimised by addition of 3% glycerol and by batch seeding. The ternary complex crystallized in the orthorhombic space group P2$_1$2$_1$2$_1$, with cell dimensions: a = 111.7 Å b = 145.7 Å c = 197.2 Å, and two copies of the ternary complex in the asymmetric unit. X-ray diffraction data were collected at beamline I02 of the Diamond Light Source, and processed as for the native Pol α crystal structure. The structure was solved by molecular replacement in PHASER, using the crystallographic model of the native Pol α as search model, and refined to 3.1 Å resolution with BUSTER and PHENIX, to

**Table 2.** Oligonucleotides used in the primer extension assay of *Figure 4D*

| Primer name | Type | Primer$_{5-3'}$ | Template$_{5-3'}$ |
|---|---|---|---|
| A12 | RNA | $A_{12}$ | $T_{50}$ |
| dA12 | DNA | $dA_{12}$ | $T_{50}$ |
| G2 | DNA | AAAAAGGAAAAA | $T_{43}CCT_5$ |
| G4 | DNA | AGGAAGGAAAAA | $T_{43}CCTTCCT$ |
| G6 | DNA | AGGAAGGAGGAA | $T_{40}CCTCCTTCCT$ |
| G7 | DNA | AGGAGGGAGGAA | $T_{40}CCTCCCTCCT$ |

R-factor/R-free (%) of 21.1/24.8. Amino acids 507–510 (4 residues), 677–680 (4 residues), 816–839 (24 residues), 1175–1187 (13 residues) and 1243–1258 (16 residues) were not visible or poorly ordered in the electron density maps during refinement and were therefore excluded from the final refined model. Details of data processing and the crystallographic refinement are provided in *Table 1*.

## Primer extension assay

The assay was performed in a 20 µl reaction containing 10 nM wild-type Pol α (amino acid 349–1258), 0.75 µM template poly(dT) DNA 70mer (Sigma-Genosys), 0.5 µM oligo(dA) DNA or oligo(A) RNA 15mer (Sigma-Genosys), 100 µM dATP in 20 mM Tris-HCl pH 8.0, 50 mM NaCl, 10 mM $MgCl_2$, 0.2 mg/ml BSA, 2 mM DTT buffer. Reactions were initiated by the addition of 1 µl of radio-labeled 2 mM dATP (1:30 v/v dilution of 0.7 µCi [$\alpha^{32}$P] dATP with cold dATP), incubated at 37°C for the appropriate time and quenched by addition of 12 µl of formamide loading buffer (95% formamide, 0.025% bromophenol blue, 0.025% xylene cyanol, 5 mM EDTA and 0.025% SDS) and heating at 70°C for 5 min. The reaction products were separated by electrophoresis in denaturing conditions, on a pre-run 7 M urea 12.5% polyacrylamide gel run for 2 hr at 2000 V. Gels were fixed in 10% acetic acid, 10% ethanol for 5 min, dried under vacuum and imaged by storage phosphor autoradiography in a Typhoon™ FLA 9000 biomolecular imager (GE Healthcare). Quantitative gel analysis was performed with ImageQuant TL (GE Healthcare).

The RNA and DNA sequences used in the experiment of *Figure 4D* are shown in *Table 2*.

Residual activity of the D998N R508A N509A Pol α mutant was demonstrated by the primer extension assay, as described; the protein concentration in the assay was varied over a range of concentrations up to 10 µM (*Figure 1—figure supplement 1*).

## Molecular dynamics

In preparation for the simulation, three loop regions of Pol α corresponding to amino acids 677–679 (3 residues), 1056–1062 (7) and 1176–1185 (10), that are disordered in the crystal structures of Pol α, were modeled using MODELLER9v8 (*Sali and Blundell, 1993*; *Fiser et al., 2000*). A longer, disordered loop region spanning amino acids 816–840 (25 residues) was too long for modeling without template; the equivalent region in the crystal structure of Pol δ (residues 578–585; PDB ID 3IAY) was used instead. The C-terminal helix of the thumb domain (residues 1229–1242) is disordered in the apo structure and was modeled based on its conformation in the structure of the ternary complex. After modelling of the missing loops, the Pol α polypeptide comprises 877 residues. In addition to the Pol α polypeptide, the ternary complex contains one template DNA molecule of 16 deoxyribonucleotides, one RNA primer of nine ribonucleotides extended at the 3'-end with two deoxynucleotides and one deoxyguanosine triphosphate (dGTP). The 5'-terminal ribonucleotide of the RNA primer was excluded from the model of ternary complex, as it makes extensive contacts with the other Pol α chain in the asymmetric unit and does not form a Watson-Crick base pair.

The dynamic behaviour of Pol α was explored by simulating the trajectory of the apo structure (APO trajectory), of the ternary complex (HOLO trajectory) and of the polymerase structure in the conformation adopted in the ternary complex but in the absence of RNA/DNA and dGTP (APOFORCED trajectory). A total of six trajectories were simulated: two APO trajectories of 150 ns, two HOLO trajectories of 100 ns and two APOFORCED trajectories of 100 ns.

Molecular dynamics simulations were performed with the AMBER11 package, using the AMBER FF99SB force field (*Wang et al., 2004*). A dodecahedral box of water molecules, treated as in the TIP3P model (*Jorgensen et al., 1983*), was built around the complexes and a physiological concentration of 0.15 M NaCl (72 counterions each) was added to the box. The dGTP molecule, present in the HOLO structure, has been parameterized using the GAFF code, as implemented in AMBER. The following protocol was used for all the simulations: (1) *in vacuo* minimization (1000 steps); (2) minimization, keeping the complexes fixed, allowing water molecules and ions to equilibrate (1000 steps of steepest descent plus 1000 steps of conjugate gradient); (3) minimization of all the system, without restrictions (1000 steps of steepest descent plus 1000 steps of conjugate gradient); (4) NVT equilibration, 1 ns; (5) production phase. All calculations were performed with the CUDA-enabled version of PMEMD (*Goetz, et al., 2012*), using TESLA GPUs at the High Performance Computing (HPC) cluster of the University of Cambridge. Two TESLA-GPUs perform approximately 4 ns/day, when computing a system of approximately 115000 atoms. Analysis of the trajectories was performed with the AMBERTOOLS 1.5 and GROMACS packages (*Hess et al., 2008*).

Principal component analysis was used to compare the principal modes of motion of Pol α in the APO, HOLO and APOFORCED trajectories. The covariance matrix of the APOFORCED trajectories was constructed, based on the three-dimensional positional fluctuations of C-α atoms from their ensemble average position, and diagonalized, creating a set of eigenvectors and eigenvalues that represent the direction and the amplitude of the motion, respectively. All the trajectories were projected on the first two eigenvectors and eigenvalues of the APOFORCED trajectories, in order to compare the regions sampled along common eigenvectors and to analyze the distribution of the motion. Construction and diagonalization of the covariance matrix was performed using the g_covar command of GROMACS. Projection of structures onto eigenvectors was performed using the g_anaeig command of GROMACS. All the other analyses have been performed with GROMACS tools, such as g_rms, g_rmsf, g_cluster. AmberTools's ptraj was used for the DSSP calculations and for all the transformation of the trajectories.

Artwork figures were prepared with PyMOL (Version 1.5.0.4, Schrödinger LLC, http://www.pymol.org/), Chimera (*Pettersen et al., 2004*) and QuteMol (*Tarini et al., 2006*). Movies were prepared with Chimera.

## Acknowledgements

We would like to thank Ludovic Sauguet for help and advice in the initial stages of the project, Dimitri Chirgadze for assistance using the X-ray crystallography facility in the Department of Biochemistry, Robert Glen for access to the computing facilities of the Unilever Centre for Molecular Informatics, Department of Chemistry, University of Cambridge, Giorgio Colombo for help and advice with the molecular dynamics simulations, Len Packman for mass spectrometry of RNA in the ternary complex crystals.

Coordinates and structure factors have been deposited with the Protein Data Bank under accession codes 4FVM (apo), 4FXD (binary complex) and 4FYD (ternary complex).

## Additional information

### Funding

| Funder | Grant reference number | Author |
| --- | --- | --- |
| Wellcome Trust | 084279/Z/07/Z | Luca Pellegrini |

The funder had no role in study design, data collection and interpretation, or the decision to submit the work for publication.

### Author contributions

RLP, RT, SK, JDM, Conception and design, Acquisition of data, Analysis and interpretation of data, Drafting or revising the article; MLK, LP, Conception and design, Analysis and interpretation of data, Drafting or revising the article

# Additional files

## Major datasets

The following datasets were generated:

| Author(s) | Year | Dataset title | Dataset ID and/or URL | Database, license, and accessibility information |
|---|---|---|---|---|
| Perera RL, Torella R, Klinge S, Kilkenny ML, Maman JD, Pellegrini L | 2012 | Crystal structure of yeast DNA polymerase alpha | 4FVM; http://www.rcsb.org/pdb/search/structidSearch.do?structureId=4FVM | Publicly available at the RCSB Protein Data Bank (http://www.rcsb.org/pdb/). |
| Perera RL, Torella R, Klinge S, Kilkenny ML, Maman JD, Pellegrini L | 2012 | Crystal structure of yeast DNA polymerase alpha bound to DNA/RNA | 4FXD; http://www.rcsb.org/pdb/search/structidSearch.do?structureId=4FXD | Publicly available at the RCSB Protein Data Bank (http://www.rcsb.org/pdb/) |
| Perera RL, Torella R, Klinge S, Kilkenny ML, Maman JD, Pellegrini L | 2012 | Crystal structure of yeast DNA polymerase alpha bound to DNA/RNA and dGTP | 4FYD; http://www.rcsb.org/pdb/search/structidSearch.do?structureId=4FYD | Publicly available at the RCSB Protein Data Bank (http://www.rcsb.org/pdb/). |

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
