## [Decision Letter]

Thank you for choosing to send your work entitled “Mechanism for priming DNA synthesis by yeast DNA Polymerase α” for consideration at *eLife*. Your article has been favorably evaluated by a Senior editor and 2 reviewers. The Reviewing editor and the other reviewer discussed their comments before we reached this decision, and the Reviewing editor has assembled the following comments to help you prepare a revised submission. We anticipate that you will be able to deal with all of these comments by revising the text of the manuscript, and we look forward to receiving your revised manuscript soon.

This study investigates one of the most frequent DNA transactions in a eukaryotic cell, the initiation of Okazaki fragment synthesis during nuclear DNA replication. The manuscript describes X-ray crystal structures of the catalytic subunit of yeast DNA polymerase alpha in three states (apo, bound to RNA-primed DNA, and a complete catalytic complex). The polymerase is shown to bind to an RNA/DNA helix containing a turn of A-form base pairs in the duplex upstream of the active site. Biochemical and computational results are presented that lead the authors to suggest a mechanism for the termination of primer synthesis by polymerase alpha after one helical turn, allowing transfer of the primer-template to the DNA polymerases that perform most of nuclear DNA replication. The enzyme binds selectively to the A-form helix embodied by the RNA-DNA substrate. Primer extension with deoxynucletides is expected to change the conformation of the primer-template towards a B-form helix, which would not be optimal for the observed contacts seen with the A-form primer-template in the crystal structure. Release of Pol α enables Pol δ to proceed to complete the faithful synthesis of Okazaki fragments.

This is an excellent paper in which the structural and biochemical analyses are augmented by molecular dynamics simulations that show that the polymerase is so constructed that it requires the presence of the RNA-DNA hybrid to maintain a closed form (in contrast to a bacteriophage DNA polymerase of the same family). The crystallographic data presented are impressive and are technically sound, and the structures are novel. The observations presented here are consistent with the authors' hypothesis about DNA release from Pol α, which is logical and elegant in its simplicity. This manuscript elucidates a fundamental mechanism of eukaryotic replication and will therefore be of general interest to a large group of biologists. Publication in *eLife* is recommended after the manuscript is revised to address the following comments.

1) A key point of emphasis in the manuscript is that Pol α synthesizes about 10 nt. of DNA (i.e., a turn of duplex) before dissociation. This is a key point because that length of DNA suffices to release the RNA from the Pol binding site. The reviewers are confused by apparently contradictory statements about the length of the DNA segment synthesized in the literature. For example, reviews by Burgers (2009) [JBC 284:4041] and Arezi and Kuchta (2000) [TIBS 25: 572] both state that ∼20 nt of DNA are synthesized. If 20 nts of DNA are synthesized, the duplex bound by DNA polymerase α would be all DNA, and the structural features that are suggested to be important for termination would no longer be relevant. We recognize that both reviews simply state this “fact” without attribution, but a review by Balakrishnan and Bambara (2011) [JBC 286:6865] clearly states that experiments with the SV40 system shows synthesis of ∼20 nucleotides of DNA by the primase. What is the length of DNA synthesized by Pol α before dissociation? The authors should clear up this point of confusion by referring clearly to the literature concerning the length of DNA synthesized by Pol α as opposed to SV40. If this is a matter of uncertainty then the manuscript should be revised accordingly to reflect the uncertainty.

2) The authors indicate that the structure of the ternary complex depicts an actively copying polymerase. The protein was pre-incubated with RNA-DNA and dGTP. Could the observed incorporation have occurred prior to crystal formation? Why is the incoming dGTP observed in the ternary complex not incorporated? Are all atoms needed for catalysis observed and in the correct geometry? If not, and/or if the resolution is insufficient, then the claim is premature. It would be very useful, and standard in this field, to shown the geometry of the polymerase active site in detail, perhaps superimposing it with that of another polymerase (e.g., RB69 Pol), to show that the primer terminus and the 3´-O is the correct position with respect to the catalytic metals and incoming nucleotide.

3) The models presented in this paper are plausible, rather than completely convincing. The model in Figure 6 might be more convincing (or not) depending on the outcome of more precise mapping experiments using a natural DNA sequence, ideally the sequence used for the crystallography, rather than the artificial template used in Figure 4. It would be straightforward to quantify termination of processive synthesis at each nucleotide position as synthesis proceeds, and map this pattern against the structural features that are hypothesized to be important for switching. This experiment could be applied to derivatives containing amino acid substitutions for residues thought to be important for switching. Even one positive result of this type would lend confidence to the model that is the main take home message of the manuscript.

It would also be great to have genetic evidence that the proposed mechanism limits genome instability, which is stated in the last sentence of the Abstract more like a fact than a possibility worth investigating. In the absence of such experiments, the authors should adjust the language to reflect the fact that they are proposing plausible but not proven models.

---

## [Author Response]

*1) A key point of emphasis in the manuscript is that Pol alpha synthesizes about 10 nt. of DNA (i.e., a turn of duplex) before dissociation. This is a key point because that length of DNA suffices to release the RNA from the Pol binding site. The reviewers are confused by apparently contradictory statements about the length of the DNA seg,ment synthesized in the literature. For example, reviews by Burgers (2009) [JBC 284:4041] and Arezi and Kuchta (2000) [TIBS 25: 572] both state that ∼20 nt of DNA are synthesized. If 20 nts of DNA are synthesized, the duplex bound by DNA polymerase alpha would be all DNA, and the structural features that are suggested to be important for termination would no longer be relevant. We recognize that both reviews simply state this “fact” without attribution, but a review by Balakrishnan and Bambara (2011) [JBC 286:6865] clearly states that experiments with the SV40 system shows synthesis of ∼20 nucleotides of DNA by the primase. What is the length of DNA synthesized by Pol α before dissociation? The authors should clear up this point of confusion by referring clearly to the literature concerning the length of DNA synthesized by Pol α as opposed to SV40. If this is a matter of uncertainty then the manuscript should be revised accordingly to reflect the uncertainty*.

We agree with the reviewers about the confusion that persists in the literature concerning the precise extent of Pol α polymerisation. As the editor points out, the reviews by Burgers (2009) [JBC 284:4041] and Arezi and Kuchta (2000) [TIBS 25: 572] do not cite primary references. Furthermore, the review by Balakrishnan and Bambara (2011) [JBC 286:6865] mentioned by the editor, refers to the same review by Arezi and Kuchta (2000) [TIBS 25: 572] (penultimate sentence of page 6865), when discussing the length of DNA synthesized by Pol α. The current state of our knowledge concerning the length of DNA polymerised by Pol α and its role of in priming nucleic acid synthesis is authoritatively summarised by De Pamphilis and Bell in their recent book “Genomic Duplication” (2011), Garland Science. Quoting from Chapter 5, page 102: “Once DNA synthesis begins, Pol α processively extends the RNA primer by approximately 10 dNMPs, and then Pol α begins to dissociate from the primer:template as it begins to incorporate the next 10 dNMPs, thereby allowing RNA-p-DNA primers to become accessible to other DNA polymerases”. The book by De Pamphilis and Bell (2011), Garland Science, is referenced in the opening paragraph of the Introduction.

Existing biochemical evidence, obtained using similar primer/template substrates to ours, is in agreement with our observation of extension products that peak stochastically at approximately 10–12 deoxynucleotides; see for instance, Singh et al, (1986) [JBC 261:8564]; Brooks and Dumas (1989) [JBC 264:3602]; Kuchta et al (1990) [JBC 265:16158]. In particular, the experiments described in Figure 7 of Brooks and Dumas, (1989) [JBC 264: 3602] allow a direct estimate of the size of the DNA segment of primer polymerised by Pol α, with a distribution of sizes that closely matches what we observe in our extension assays. All of these papers are referenced in the relevant part of the Introduction.

In the same page of their book, DePamphilis and Bell make the prescient comment that:

“Transition from an A-form to a B-form duplex may also trigger the hand-off from Pol α either to Pol δ on the lagging strand template or to Pol ε on the leading strand template by stimulating release of Pol α from the primer:template”.

Our data provide a structural basis for the mechanism of RNA primer extension and release by Pol α envisaged by the authors.

*2) The authors indicate that the structure of the ternary complex depicts an actively copying polymerase. The protein was pre-incubated with RNA-DNA and dGTP. Could the observed incorporation have occurred prior to crystal formation? Why is the incoming dGTP observed in the ternary complex not incorporated? Are all atoms needed for catalysis observed and in the correct geometry? If not, and/or if the resolution is insufficient, then the claim is premature. It would be very useful, and standard in this field, to shown the geometry of the polymerase active site in detail, perhaps superimposing it with that of another polymerase (e.g., RB69 Pol), to show that the primer terminus and the 3´-O is the correct position with respect to the catalytic metals and incoming nucleotide*.

We believe that polymerization by the D998N Pol α mutant happens during vapour-diffusion crystallization, in the time interval before nucleation of the crystal lattice. Incorporation of two deoxynucleotides, followed by translocation onto the next templating base (position 0 in Figure 2C) and binding of a third dGTP molecule is required in order to trigger formation of the crystal lattice, capturing Pol α in an actively copying conformation. Figure 1–figure supplements 6 and 7 show the non-crystallographic interaction between one polymerase molecule and the DNA/RNA duplex of the NCS-related complex, which is responsible for trapping Pol α in the crystal lattice and arresting further polymerisation.

In order to seek confirmation of our hypothesis, we devised a crystallisation experiment with an RNA primer/ DNA template substrate that contained an RNA 9mer, i.e., one base shorter of the primer used in the original crystallization experiment. We reasoned that, if addition of two dGTPs to the RNA 10mer in the crystallisation drop was necessary in order to obtain a RNA/DNA duplex of the required length (10 + 2 = 12 bp) to promote crystal growth, the same crystal form could be produced by extension of an RNA 9mer with three dGTPs (9 + 3 = 12 bp). Indeed, the crystallisation experiment with the shorter RNA primer was successful and we were able to solve the structure of the active complex, albeit at the lower resolution of 3.6 Å (Perera and Pellegrini, unpublished data). As predicted, the new crystal structure had the same unit cell, symmetry and non- crystallographic packing of the original structure.

The active site of Pol α is shown in Figure 1–figure supplement 4. We have now updated the figure as suggested by the reviewer, so that it shows the active site of Pol α superimposed on that of yeast Pol δ (PDB id: 3IAY), its closest structural homologue in the PDB. The superposition shows that the geometry of Pol α's active site is very closely related to that of Pol δ; therefore, we believe that nucleotide polymerization by Pol α proceeds via the same two-metal ion catalysis demonstrated previously for other DNA polymerases. Due to the limited resolution of our study, we did not include Mg2+ ions in our crystallographic model.

*3) The models presented in this paper are plausible, rather than completely convincing. The model in Figure 6 might be more convincing (or not) depending on the outcome of more precise mapping experiments using a natural DNA sequence, ideally the sequence used for the crystallography, rather than the artificial template used in Figure 4. It would be straightforward to quantify termination of processive synthesis at each nucleotide position as synthesis proceeds, and map this pattern against the structural features that are hypothesized to be important for switching. This experiment could be applied to derivatives containing amino acid substitutions for residues thought to be important for switching. Even one positive result of this type would lend confidence to the model that is the main take home message of the manuscript*.

*It would also be great to have genetic evidence that the proposed mechanism limits genome instability, which is stated in the last sentence of the Abstract more like a fact than a possibility worth investigating. In the absence of such experiments, the authors should adjust the language to reflect the fact that they are proposing plausible but not proven models*.

We believe that the quantification of Pol α's extension products shown in Figure 4B effectively achieves the aim of the experiment proposed by the reviewers. We agree with the reviewers that in principle it might be informative to try to assess the relative contribution of single residues at the protein-nucleic acid interface to the specific interaction with the templated RNA primer. However, we believe that it would be hard or impossible to probe the specificity of the interaction by targeted mutagenesis without concurrently impairing the general affinity of the polymerase for its primer/template substrate. Furthermore, a genetic analysis such as the one proposed by the reviewer would be interesting, but we believe that it is beyond the scope of the current manuscript.

We have reworded the concluding remarks of the Abstract and the Discussion, in order to reflect the comments of the reviewer concerning the plausibility of the proposed mechanism of primer termination and release by Pol α.